# In Situ Growth Behavior of SiC Whiskers with High Aspect Ratio in the Synthesis of ZrB_2_-SiC Composite Powders

**DOI:** 10.3390/ma13163502

**Published:** 2020-08-08

**Authors:** Xiaoqing Lian, Xiaohu Hua, Xiaogang Wang, Lirong Deng

**Affiliations:** School of Materials Science and Engineering, Xi’an University of Science and Technology, Xi’an 710054, China; huaxiaohu@xust.edu.cn (X.H.); wangxiaogang@xust.edu.cn (X.W.); denglirong@xust.edu.cn (L.D.)

**Keywords:** in situ synthesis, whisker, SiC, ZrB_2_

## Abstract

Aiming to provide key materials in order to improve the fracture toughness of ZrB_2_ ceramics, ZrB_2_-SiC composite powders with in situ grown SiC whiskers were successfully synthesized via a simple molten-salt-assisted ferrous-catalyzed carbothermal reduction method. Thermodynamic calculations on the ZrO_2_-SiO_2_-B_2_O_3_-C-Fe system were carried out. The effects of heating temperature and ferrous catalyst amount on the growth behavior of SiC whiskers in ZrB_2_-SiC composite powders were investigated using X-ray diffraction (XRD), scanning electron microscopy (SEM), X-ray energy dispersive spectroscopy (EDS), and transmission electron microscopy (TEM). The results showed that the aspect ratio of SiC whiskers and the relative content of ZrB_2_ particles increased with increasing heating temperature (1523–1723 K) and a molar ratio of Fe to ZrSiO_4_ from 0:1 to 0.2:1. Phase-pure ZrB_2_-SiC composite powders were obtained at 1723 K when the molar ratio of raw materials was 0.2:0.5:1:1.5:8.4 (Fe:NaCl:ZrSiO_4_:B_2_O_3_:C). Single crystalline β-SiC whiskers with a mean diameter of 0.15 μm and an aspect ratio of 70–120 were homogeneously distributed in the final composite powders. A molten-salt-assisted iron-catalyzed vapor–solid mechanism was promoted for the growth mechanism of in situ grown SiC whiskers.

## 1. Introduction

Owing to their unique advantages including low density, high melting point, and stable chemical properties, ZrB_2_ ultrahigh-temperature ceramics are promising materials for leading edges and propulsion components of hypersonic aerospace vehicles and advanced reusable atmospheric reentry vehicles [1,2,3]. However, they suffer from difficult densification, poor oxidation resistance, and low fracture toughness [4]. Incorporating SiC whiskers into the ZrB_2_ ceramic matrix was found to be effective in improving these performances, especially for fracture toughness [5,6,7]. For example, fracture toughness of monolithic ZrB_2_ ceramics was commonly only 3–4 MPa·m^1/2^ [8], while that increased to above 5 MPa·m^1/2^ through a bridging toughening mechanism if 20 vol % SiC whiskers were added [6]. However, except for the inevitable grinding damage, it was sometimes difficult to obtain the uniform dispersion of SiC whiskers, which limited the improvement of the performance of ZrB_2_-SiC composite ceramics. An effective way to overcome the problems is to synthesize ZrB_2_-SiC composite powders with in situ grown SiC whiskers.

ZrB_2_-SiC composite powders have been synthesized through various approaches [9,10,11,12,13,14]. Xie [10] synthesized spherical ZrB_2_-SiC composite powders ranging from 100 nm to 300 nm through a one-step reduction process of ZrO_2_, B_4_C, carbon black, and silicon. Li [11] prepared ZrB_2_-SiC composite powders by molten-salt-mediated reduction of ZrSiO_4_, B_2_O_3_, activated carbon, and Mg, which exhibited grain sizes of several microns and comprised SiC nanoparticles well distributed in the ZrB_2_ matrix. ZrB_2_-SiC composite powders were synthesized by Cao [12] via a combined sol–gel and microwave boro/carbothermal reduction process using zirconium oxychloride, boric acid, tetraethoxysilane, and glucose as starting materials; the resulting crystalline sizes of ZrB_2_ and SiC were about 58 and 27 nm, respectively. The as-prepared powders thus obtained were with isometric morphology. Some studies on anisotropic ZrB_2_-SiC composite powders were reported in recent years. Lin [15] prepared ZrB_2_-SiC composite powders with in situ rod-shaped ZrB_2_. As-prepared ZrB_2_-SiC composite powders were modified by Zhong [16] using in situ grown SiC nanowires with a diameter of 200 nm. However, studies on the preparation of ZrB_2_-SiC composite powders with in situ grown SiC whiskers have not been as widely reported.

SiC whiskers or nanowires were commonly prepared by molten-salt-assisted (using NaCl, KCl, or NaF) [17] or transition-metal-catalyzed (using ferrous, cobalt, or nickel) carbothermal reduction of silicon dioxide (SiO_2_) [18]. In this work, a simple molten-salt-assisted iron-catalyzed carbothermal reduction method was employed to synthesize ZrB_2_-SiC composite powders with in situ grown SiC whiskers using zircon, boron oxide, carbon black, ferrous powders, and NaCl as raw materials. Aiming to obtain SiC whiskers with ideal morphology (straight shape, small diameter, high aspect ratio, etc.) and clarify their in situ growth behavior in ZrB_2_-SiC composite powders, based on the thermodynamic calculations on the ZrO_2_-SiO_2_-B_2_O_3_-C-Fe system, the effects of heating temperature and ferrous catalyst amount on the phase and microstructure of as-prepared powders, especially on the morphology evolution of SiC whiskers were investigated. Further, growth mechanisms of in situ grown SiC whiskers were investigated. This work may serve as a theoretical basis for the preparation of anisotropic ZrB_2_-SiC composite powders.

## 2. Materials and Methods

Commercially available powders of zircon (ZrSiO_4_, 97.20%, ~14.3 μm, Chenyuan Powder, Zibo, Shandong, China), boron oxide (B_2_O_3_, 99.9999%, ~2.2 μm, Xingye Metal, Xingtai, Hebei, China), carbon black (C, 99.50%, ~3.6 μm, Meidi Family, Shanghai, China), ferrous metal s (Fe, 99.9999%, ~0.2 μm, Chengxin Metal, Qinghe, Hebei, China), and NaCl (99.50%, ~3.6 μm, Guoyao Chemical, Shanghai, China) were used as raw materials to synthesize ZrB_2_-SiC composite powders according to the following reaction:ZrSiO_4_ (s) + B_2_O_3_ (l) +7C (s) = ZrB_2_ (s) +SiC (s) +7CO (g)(1)

Molar ratios of raw materials are listed in Table 1. Excess boron oxide and carbon black were employed with a molar ratio of 1:1.5:8.4 (ZrSiO_4_:B_2_O_3_:C). A small amount of NaCl was added with a molar ratio of 0.5:1 (NaCl:ZrSiO_4_). Samples with different ferrous catalyst amounts with molar ratios of 0:1, 0.1:1, 0.2:1, 0.3:1, and 0.4:1 (Fe:ZrSiO_4_) were designed, and the corresponding products were referred to as ZS0, ZS1, ZS2, ZS3, and ZS4.

After dry mixing for 2 h, the mixed powders were put into a corundum crucible and heated at temperatures ranging from 1573 to 1773 K for 3 h under flowing argon gas (99.999%, 0.5 L/min) in a box atmosphere furnace. The heating rate was 10 K/min. The as-prepared powders were washed repeatedly with hot distilled water to remove residual salt and dried at 383 K for 24 h.

Thermodynamic calculations on the ZrO_2_-SiO_2_-B_2_O_3_-C-Fe system were performed using the Enthalpy–Entropy–Heat Capacity (HSC) Chemistry 6.0 software. The phase composition was characterized by X-ray diffraction (XRD, Shimadzu XRD-7000, Kyoto, Japan) with Cu-K*_α_*_1_ radiation (*λ* = 0.154 nm) at a scan rate of 5° (2*θ*)/min. The relative contents of crystalline phase were calculated by HighScore Plus software. Scanning electron microscopy (SEM, JSM-6390A, Kyoto, Japan) along with X-ray energy-dispersive spectroscopy (EDS) were used to analyze the morphology of composite powders and assist in phase identification, respectively. Transmission electron microscopy (TEM, JEM-2100, Kyoto, Japan) along with high-resolution transmission electron microscopy (HRTEM) and selected area electron diffraction (SAED) were used to analyze the morphology of SiC whiskers.

## 3. Results and Discussion

### 3.1. Thermodynamic Calculation

Thermodynamic calculation of the ZrO_2_-SiO_2_-B_2_O_3_-C-Fe system was performed to predict the reaction production at different temperatures. It is known that pure ZrSiO_4_ starts decomposing into monoclinic ZrO_2_ (*m*-ZrO_2_) and amorphous SiO_2_ at a temperature above 1725 K [19]; however, this temperature may be practically lower because of the existence of NaCl molten salt and liquid B_2_O_3_ and the formation of low-temperature eutectic melts from impurities of zircon [20].

Main possible reactions in this work and their Gibbs free energies ($\Delta_{r}G$) at a temperature range of 1000–2000 K are presented in Table 2. Standard Gibbs free energies (${\;\Delta }_{r}G_{}^{\theta }$) of Reactions (2)–(6) at different temperatures were calculated using the HSC Chemistry 6.0 software, and CO partial pressure ($p_{co}$) was estimated to be equal to standard atmospheric pressure ($\;p^{\theta }$) during the reaction process under flowing argon gas; thus, the corresponding Gibbs free energy ($\Delta_{r}G$) results were obtained, as shown in Figure 1. Setting the Gibbs free energy to be zero, it was calculated that Reactions (2)–(6) could occur at 1776, 1793, 1940, 1503, and 1588 K, respectively, leading to the appearance of new phases ZrB_2_, SiC, ZrC, Fe_2_B, and FeB, respectively.
ZrO_2_ (s) + B_2_O_3_ (l) + 5C (s) = ZrB_2_ (s) + 5CO (g)(2)
SiO_2_ (s) + 3C (s) = SiC (s) + 2CO (g)(3)
ZrO_2_ (s) + 3C (s) = ZrC (s) + 2CO (g)(4)
4Fe (l)+B_2_O_3_ (l)+3C (s) = 2Fe_2_B (s)+3CO (g)(5)
2Fe (l)+B_2_O_3_ (l)+3C (s) = 2FeB (s)+3CO (g)(6)

### 3.2. Phase Composition Evolution of Composite Powders

To determine the lowest heating temperature needed to obtain phase-pure ZrB_2_-SiC composite powders, the XRD patterns and the relative contents of crystalline phase of sample ZS2 heated at temperatures ranging from 1573 to 1773 K for 3 h were studied, as shown in Figure 2.

Diffraction peaks of ZrB_2_ and *β*-SiC were observed at 1573 K, while no diffraction peaks of ZrSiO_4_ were detected (Figure 2a), indicating that ZrB_2_ and *β*-SiC had begun to nucleate and grow before 1573 K through Reactions (2) and (3), which was much lower than the thermodynamic temperatures of 1776 and 1793 K as shown in Figure 1. When increasing the temperature from 1573 to 1723 K, ZrB_2_ content increased continuously from 16 to 72% (Figure 2b). The slope of the line for Reaction (2) was negative in Figure 1, so increasing the temperature was helpful in forming ZrB_2_. Meanwhile, *β*-SiC content changed little, indicating that formation process of *β*-SiC through Reaction (3) had completed at a temperature lower than 1573 K, which was 150 K lower than that of Zhang [17] via molten-salt-assisted carbothermal reduction with silica fume and phenolic resin as raw materials and close to that of Guo [21] via nickel-catalyzed carbothermal reduction of polycarbosilane (PCS). The relative contents of ZrB_2_ and *β*-SiC showed no obvious changes when further raising the temperature to 1773 K. The weight ratio of ZrB_2_ to *β*-SiC was 72:28 (Figure 2b), which was close to the theoretical value of 74:26 calculated based on Reaction (1), suggesting that phase-pure ZrB_2_-SiC composite powders were synthesized at 1723 K. This temperature was 50 K lower than that in the traditional carbothermal reduction of zircon [9]. Therefore, the lowest heating temperature to obtain phase-pure ZrB_2_-SiC composite powders was set as 1723 K.

Ferrous catalyst amount is another important factor influencing the phase composition of as-prepared composite powders. Figure 3 shows the XRD patterns and the relative contents of crystalline phase of samples ZS0–ZS4 heated at 1723 K for 3 h. With increasing molar ratio of Fe to ZrSiO_4_, ZrB_2_ peaks were increasingly strong (Figure 3a), and its relative contents increased significantly (Figure 3b). However, the relative contents of *β*-SiC changed little (Figure 3b), indicating that the effects of ferrous catalyst on promoting formation of *β*-SiC as reported in previous work [22] were negligible at a temperature as high as 1723 K. Hence, phase-pure ZrB_2_-SiC composite powders were synthesized when the molar ratio of Fe to ZrSiO_4_ was 0.2:1, in which the weight ratio of ZrB_2_ to *β*-SiC was 72:28 (Figure 3b). Further increasing the molar ratio of Fe to ZrSiO_4_, ZrC peaks appeared (Figure 3a). Therefore, the optimum molar ratio of raw materials to synthesize phase-pure ZrB_2_-SiC composite powders was 0.2:0.5:1:1.5:8.4 (Fe:NaCl:ZrSiO_4_:B_2_O_3_:C), i.e., sample ZS2.

### 3.3. Microstructure of ZrB_2_-SiC Composite Powders with In Situ Grown SiC Whiskers

The microstructure of sample ZS2 heated at 1723 K for 3 h was examined using SEM and EDS, as shown in Figure 4.

There are mainly three morphologies shown in Figure 4a, including whiskers with a mean diameter of 0.15 μm and aspect ratio of 70–120, tabular grains with a diameter of 4–5 μm, and nanoparticles with a mean diameter of 0.03 μm. The EDS spectra at point 001 showed the existence of elements Si and C (Figure 4b), and the EDS spectra at point 002 showed the existence of elements Zr and B (Figure 4c), so the whiskers were SiC and tabular grains were ZrB_2_, which was confirmed by EDS mapping of the Si element (Figure 4d) and Zr element (Figure 4e) as well. Meanwhile, it is shown in Figure 4d that the nanoparticles in Figure 4a were SiC. It was seen that SiC whiskers were homogeneously distributed in the final composite powders.

There were minor spectra of Fe element at typical point 001, but no spectra of B element were seen, confirming that Fe element existed as neither Fe_2_B nor FeB, i.e., Reactions (5) and (6) did not occur owing to some dynamical factors, although thermodynamic temperatures of Fe_2_B and FeB (1503 and 1588 K) were lower than that of SiC (1776 K). Neither Fe_2_B nor FeB was found in the ZrB_2_ whiskers prepared by Khanr [23] using ZrO_2_, H_3_BO_3_, C, NaCl, and catalyst Ni/Co/Fe as raw materials, which was consistent with the results of this work.

The growth direction and crystal structure of SiC whiskers were further investigated by TEM, HRTEM, and SAED, as seen in Figure 5. The single SiC whisker was smooth and straight with a diameter of about 0.15 μm (Figure 5a). The lattice space value was measured to be 0.25 nm (Figure 5b), which was in agreement with the plane distance of the (111) plane [24], the close-packed planes of *β*-SiC, and so it was concluded that the whiskers preferentially grew along the <111> direction. The SAED patterns confirmed that the SiC whisker was single crystalline *β*-SiC (3C-SiC).

### 3.4. Morphology Evolution of SiC Whiskers

The morphology of SiC whiskers plays an important role in toughening mechanisms, including crack bridging, crack deflection, and pullout effects [5,6,7], and thus the morphology evolution of SiC whiskers was studied by comparing the microstructures of samples at different heating temperatures and ferrous catalyst amounts, as shown in Figure 6. Table 3 lists the morphology parameters of SiC whiskers in the composite powders.

With increasing heating temperature from 1623 to 1723 K (Figure 6a,d) and increasing molar ratio of Fe to ZrSiO_4_ from 0:1 to 0.2:1 (Figure 6b–d), the aspect ratio of SiC whiskers increased while the diameter changed little. Since the relative contents of *β*-SiC changed little according to XRD analysis in Figure 2; Figure 3, it was concluded that, in the processing conditions above, the dominant growth behavior of SiC whiskers was one-dimensional growth along <111> rather than nucleation. When the molar ratio of Fe to ZrSiO_4_ increased to 0.3:1, the morphology of SiC whiskers changed from straight and fine to bent and coarse. The whiskers made contact with each other under too many catalyst droplets, hence they could not grow continuously and evenly. Furthermore, when the molar ratio rose to 0.4:1, only a few short rods remained and serious grain agglomeration occurred. Therefore, ideal morphology of SiC whiskers was obtained when the molar ratio of Fe to ZrSiO_4_ was 0.2:1 and heating temperature was 1723 K, which were the optimum processing conditions for the synthesis of phase-pure ZrB_2_-SiC composite powders as well.

Some main stepwise reactions in Reaction (3) are listed as follows [24,25,26]:SiO_2_ (s) +C (s) = SiO (g) + CO (g)(7)
SiO (g) + 3CO (g) = SiC (s) + 2CO_2_ (g)(8)
SiO (g) + 2C (s) = SiC (s) + CO (g)(9)
C (s) + O_2_ (g) = 2CO (g)(10)

Reaction (8) is the main reaction for nucleation of SiC whiskers, while Reaction (9) is the main reaction for nucleation of SiC particles. The morphology of SiC whiskers was determined by two important processes: (1) When the concentrations of SiO and CO released from Reactions (7) and (10) reached an appropriate range, SiC began to nucleate as a stable phase to reduce the system energy [27,28]. Reaction (10) occurred at a relatively higher temperature than Reaction (7) [26]. (2) With the continual provision of SiO gas and CO gas, SiC whiskers grew longer through Reaction (8) along a certain direction due to the lowest surface energy of *β*-SiC. It was concluded that high reaction rates of Reactions (7) and (8) were essential to obtain SiC whiskers with high aspect ratio.

The reactivity of SiO_2_, C, SiO gas, and CO gas were proportional to the reaction rate of Reactions (7) and (8). Based on the first-principles calculations, Wang [22] reported that the bond length in Si-O at the (101) plane of SiO_2_ became longer after adsorption of ferrous nanoparticles, indicating that iron could catalyze Reaction (7) by increasing the reactivity of SiO_2_. Wang [29] reported that the large distortion and strong interaction of nickel nanoparticles with C, SiO gas, and CO gas could promote their dissociation of atomic bonds, further making them more reactive. This conclusion could also explain the catalytic effects of ferrous powders in this work. Hence, the reaction rates of Reactions (7) and (8) were both promoted by the ferrous catalyst. A greater ferrous catalyst amount provided more active sites. The negative slope of Reaction (3) in Figure 1 also meant that increasing heating temperature effectively improved the reaction rate. That is why the aspect ratio of SiC whiskers increased with increasing ferrous catalyst amount and heating temperature, as shown in Figure 6a–d. In addition, the reaction rate of Reaction (9) was also promoted by ferrous catalyst.

The formation process of *β*-SiC through Reaction (3) was completed at 1573 K for sample Z2, as shown in Figure 2, which was due to the combined effects of the ferrous catalyst and NaCl molten salt [30], as well as high activity of amorphous SiO_2_ decomposed from ZrSiO_4_. For Reactions (2) and (4), increasing the ferrous catalyst amount was helpful in dissociating atomic bonds in carbon black and the further accelerating formation of ZrB_2_ and ZrC, as shown in Figure 3.

### 3.5. Growth Mechanisms of SiC Whiskers in ZrB_2_-SiC Composite Powders

The vapor–liquid–solid (VLS) mechanism was usually employed to explain the synthesis process of SiC whiskers with the addition of transition-metal catalysts [31,32]. The SiO gas and CO gas were absorbed into the catalyst droplet, and then SiC nucleated and grew along a direction on the interface between the nucleus and droplet. Lastly the catalyst droplets were left on the top of whiskers. However, there were no droplets found at the tips of whiskers in this work, suggesting that VLS growth mechanisms were not suitable to explain their growth behavior. Based on the XRD, SEM, EDS, TEM, HRTEM, and SAED analysis in Figure 2, Figure 3, Figure 4, Figure 5 and Figure 6, a molten-salt-assisted iron-catalyzed vapor–solid (VS) mechanism is promoted in Figure 7 and described in detail as follows:(1)Preparation stage. At a low temperature, B_2_O_3_ and NaCl melted. They accelerated the decomposition of ZrSiO_4_ into *m*-ZrO_2_ and amorphous SiO_2_ by promoting the mass transfer of raw materials.(2)Nucleation stage. Ferrous particles catalyzed Reactions (7) and (8) by increasing the reactivity of SiO_2_, C, SiO gas, and CO gas, resulting in the nucleation of SiC whiskers at a certain temperature lower than 1573 K. Meanwhile, carbon black with high reactivity was beneficial to the nucleation of ZrB_2_ grains through Reaction (2). The effect of NaCl molten salt was not evident here because the temperature approached its boiling point.(3)Growth stage. SiC whiskers grew along <111> on the solid surface of the SiC nucleus through Reaction (8) on the condition of appropriate heating temperature and ferrous catalyst amount. Some SiC nuclei which nucleated on the carbon black particles through Reaction (9) were more likely to grow into SiC nanoparticles. ZrB_2_ tabular grains continued to nucleate and grow.

## 4. Conclusions

Phase-pure ZrB_2_-SiC composite powders with in situ grown SiC whiskers were successfully synthesized by molten-salt-assisted iron-catalyzed carbothermal reduction, which has great potential application in improving the fracture toughness of ZrB_2_-SiC composite ceramics. The in situ growth behavior of SiC whiskers was studied. Conclusions are summarized as follows:(1)Phase-pure ZrB_2_-SiC composite powders were obtained when the molten ratio of raw materials was 0.2:0.5:1:1.5:8.4 (Fe:NaCl:ZrSiO_4_:B_2_O_3_:C) and the heating temperature was 1723 K, in which *β*-SiC whiskers were single crystalline with a mean diameter of 0.15 μm and aspect ratio of 70–120.(2)Heating temperature and ferrous catalyst amount obviously influenced the phase composition and microstructure of ZrB_2_-SiC composite powders, especially for the morphology of SiC whiskers. With increasing heating temperature (1523–1723 K) and molar ratio of Fe to ZrSiO_4_ (0:1 to 0.2:1), the aspect ratio of SiC whiskers increased significantly while the relative content of SiC phase changed little, and the relative content of ZrB_2_ phase increased continuously. Excess ferrous catalysts resulted in the formation of ZrC phase and serious grain agglomeration at 1723 K.(3)Molten-salt-assisted iron-catalyzed vapor–solid mechanism was promoted for the growth mechanism of in situ grown SiC whiskers in ZrB_2_-SiC composite powders. Ferrous catalysts played a major role in increasing the reactivity of SiO_2_, C, SiO gas, and CO gas, further realized by the low temperature nucleation and high aspect ratio of SiC whiskers.

Excess carbon black in the ZrB2-SiC composite powders prepared by molten-salt-assisted iron-catalyzed carbothermal reduction method was difficult to remove, which calls for further studies to increase the purity of composite powders. The later work should focus on the following two points:(1)Growth behavior of ZrB_2_-SiC composite powders with different molar ratios of ZrO_2_ to SiO_2_;(2)Evaluation of toughening effects of in situ grown SiC whiskers on ZrB_2_-SiC composite ceramics, as well as the corresponding toughening mechanisms.

## Figures and Tables

**Figure 1 materials-13-03502-f001:**
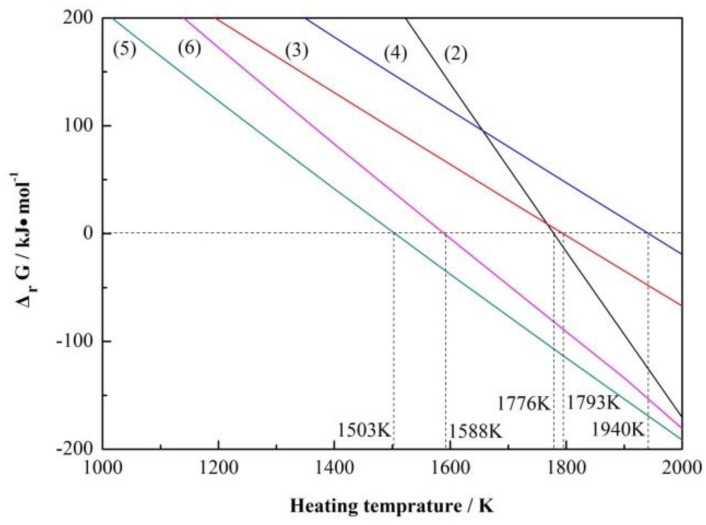
Gibbs free energies ($\Delta_{r}G$) of Reactions (2)–(6) at different temperatures.

**Figure 2 materials-13-03502-f002:**
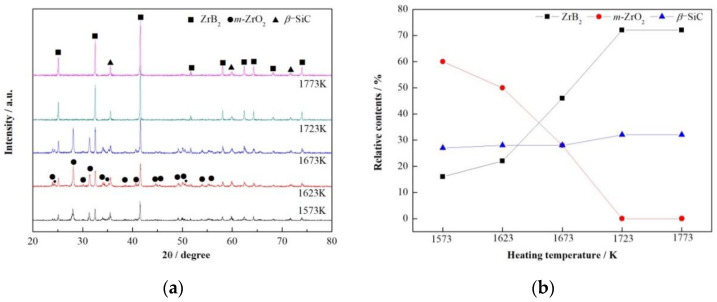
Phase composition of sample ZS2 heated at 1573–1773 K for 3 h, *n*(Fe):*n*(NaCl):*n*(ZrSiO_4_):*n*(B_2_O_3_):*n*(C) = 0.2:0.5:1:1.5:8.4. (**a**) XRD patterns; (**b**) relative contents of the crystalline phase.

**Figure 3 materials-13-03502-f003:**
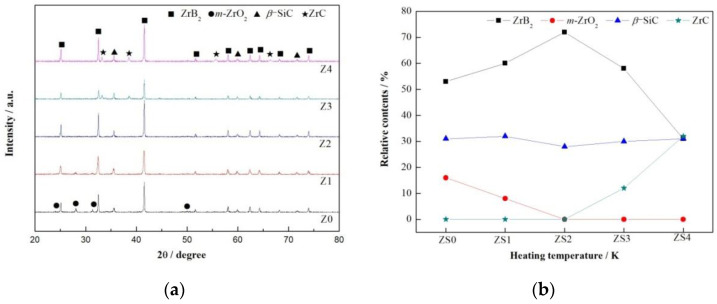
Phase composition of samples ZS0–ZS4 heated at 1723 K for 3 h, *n*(Fe):*n*(ZrSiO_4_) = 0:1, 0.1:1, 0.2:1, 0.3:1, and 0.4:1, respectively: (**a**) XRD patterns; (**b**) relative contents of the crystalline phase.

**Figure 4 materials-13-03502-f004:**
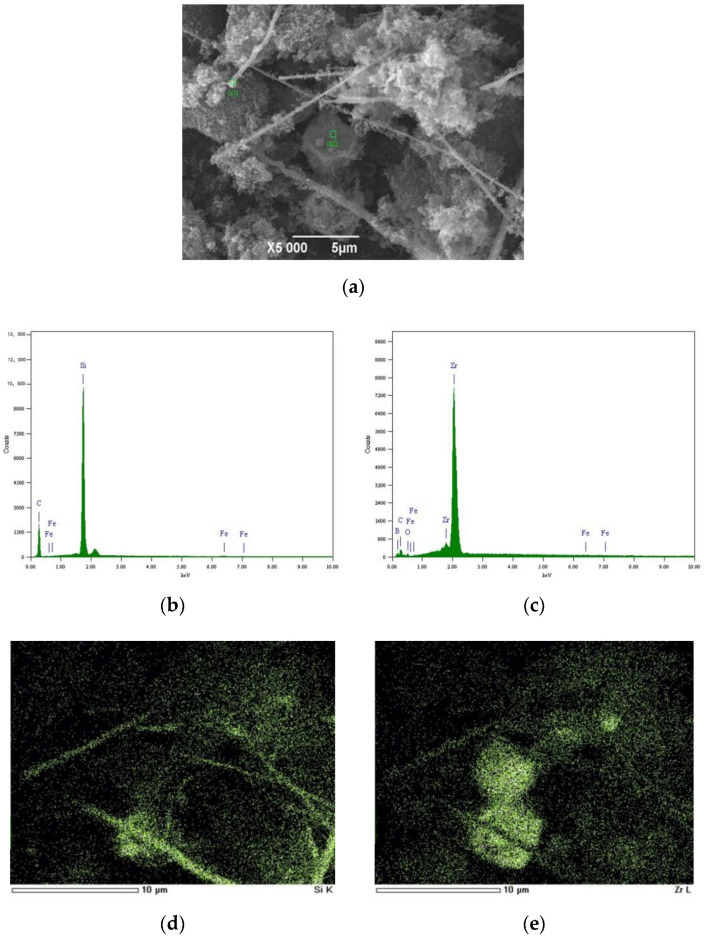
Microstructure of sample ZS2 heated at 1723 K for 3 h: (**a**) scanning electron microscopy (SEM) image; (**b**) energy-dispersive spectroscopy (EDS) spectra at point 001; (**c**) EDS spectra at point 002; (**d**) EDS mapping of Si element; (**e**) EDS mapping of Zr element.

**Figure 5 materials-13-03502-f005:**
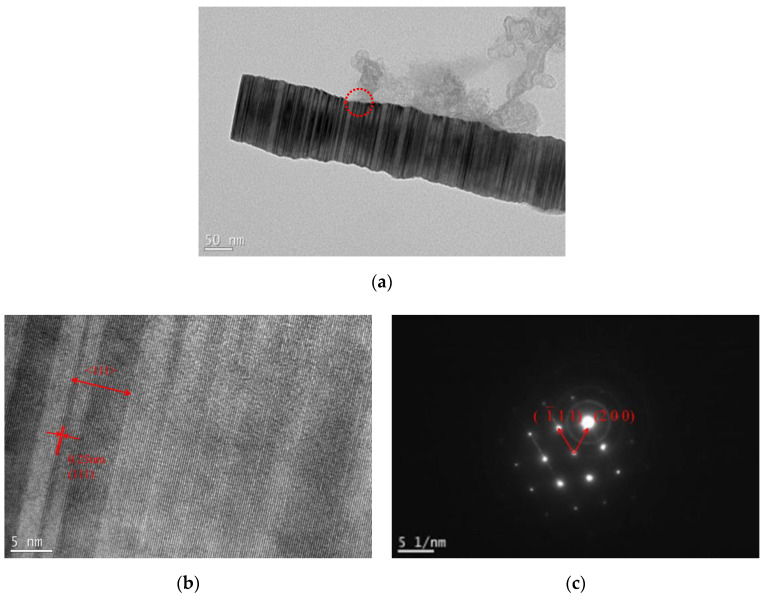
Microstructure of a single SiC whisker in sample ZS2: (**a**) transmission electron microscopy (TEM) image; (**b**) high-resolution transmission electron microscopy (HRTEM image); (**c**) selected area electron diffraction (SAED) image.

**Figure 6 materials-13-03502-f006:**
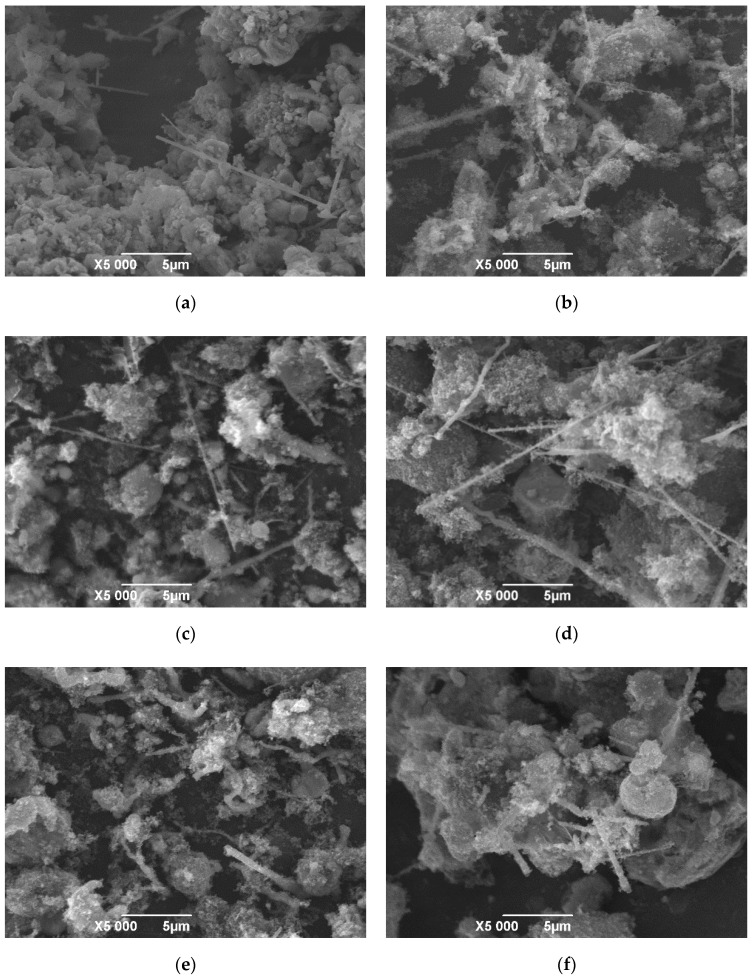
SEM images of samples at different processing conditions: (**a**) 1623 K, n(Fe):n(ZrSiO_4_) = 0.2:1; (**b**) 1723 K, n(Fe):n(ZrSiO_4_) = 0:1; (**c**) 1723 K, n(Fe):n(ZrSiO_4_) = 0.1:1; (**d**) 1723 K, n(Fe):n(ZrSiO_4_) = 0.2:1; (**e**) 1723 K, n(Fe):n(ZrSiO_4_) = 0.3:1; (f) 1723 K, n(Fe):n(ZrSiO_4_) = 0.4:1.

**Figure 7 materials-13-03502-f007:**
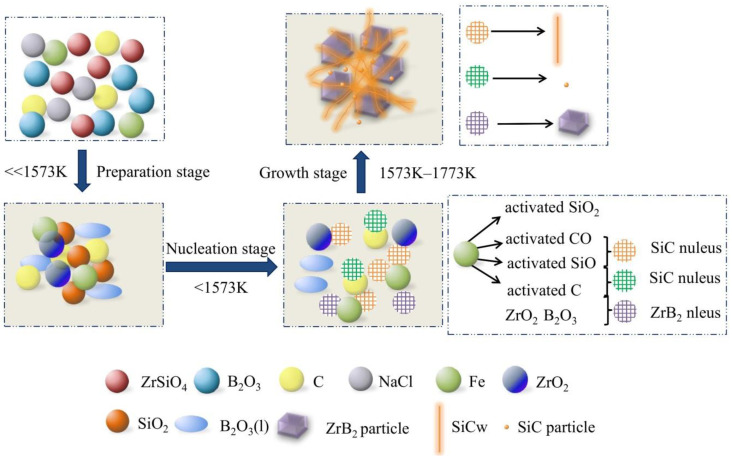
Illustrations of the possible growth process of SiC whiskers in the synthesis of ZrB_2_-SiC composite powders.

**Table 1 materials-13-03502-t001:** Molar ratios of raw materials.

Sample No.	Molar Ratios of Raw Materials				
	ZrSiO_4_	B_2_O_3_	C	Fe	NaCl
ZS0	1	1.5	8.4	0	0.5
ZS1	1	1.5	8.4	0.1	0.5
ZS2	1	1.5	8.4	0.2	0.5
ZS3	1	1.5	8.4	0.3	0.5
ZS4	1	1.5	8.4	0.4	0.5

**Table 2 materials-13-03502-t002:** Main possible reactions and their Gibbs free energies ($\Delta_{r}G$).

Reactions	$\Delta_{r}G\;$/KJ·mol^−1^
(2)	$\Delta_{r}G_{2}=\Delta_{r}G_{2}^{\theta }+5RT\ln\frac{p_{co}}{p^{\theta }}\;(\Delta_{r}G_{2}^{\theta }=-170\sim 621)$
(3)	$\Delta_{r}G_{3}=\Delta_{r}G_{3}^{\theta }+2RT\ln\frac{p_{co}}{p^{\theta }}\;(\Delta_{r}G_{3}^{\theta }=-67\sim 266)$
(4)	$\Delta_{r}G_{4}=\Delta_{r}G_{4}^{\theta }+2RT\ln\frac{p_{co}}{p^{\theta }}\;(\Delta_{r}G_{4}^{\theta }=-19\sim 322)$
(5)	$\Delta_{r}G_{5}=\Delta_{r}G_{5}^{\theta }+3RT\ln\frac{p_{co}}{p^{\theta }}\;(\Delta_{r}G_{5}^{\theta }=-191\sim 207)$
(6)	$\Delta_{r}G_{6}=\Delta_{r}G_{6}^{\theta }+3RT\ln\frac{p_{co}}{p^{\theta }}\;(\Delta_{r}G_{6}^{\theta }=-180\sim 264)$

**Table 3 materials-13-03502-t003:** Morphology parameters of SiC whiskers in samples as shown in Figure 6.

Sample No.	n(Fe):n(ZrSiO_4_)	Temperature/K	Morphology Parameters		
			Shape	Mean Diameter/μm	Aspect Ratio
Z2	0.2:1	1623 K	straight whisker	0.15	20–40
Z0	0:1	1723 K	straight whisker	0.1–0.3	20–40
Z1	0.1:1	1723 K	straight whisker	0.15	40–50
Z2	0.2:1	1723 K	straight whisker	0.15	70–120
Z3	0.3:1	1723 K	bent whisker	0.3	5–10
Z4	0.4:1	1723 K	short rod	0.3	<5

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
