# Peer review of "In Situ Growth Behavior of SiC Whiskers with High Aspect Ratio in the Synthesis of ZrB2-SiC Composite Powders"

_materials, 2020, doi:10.3390/ma13163502_

Round 1

Reviewer 1 Report

The manuscript addresses important aspects of the synthesis of ZrB2-SiC composite powders.  The manuscript is well-written, the study is complete and the references are appropriate.

However, it would help for the authors to answer the following question:

The authors are working on the subject area because of their interest in improving the fracture toughness of ZrB2 ceramics. Have the authors done any studies on improved fracture toughness with the ZrB2-SiC composite powders?  If yes, it would help to include results on this key goal of this study.

Reviewer 2 Report

The study aims to synthesize ZrB2-SiC composite powders used to improve the fracture toughness of ZrB2 ceramics. For this objective, the synthesis was carried out under different temperature and ratio of Fe and ZrSiO4. The obtained composite powders were characterized in several ways such as SEM/EDS, TEM, XRD, etc. The manuscript is interesting and can benefit the fabricating of ZrB2-SiC composite powders. The manuscript can be accepted with minor revision.

Strength:

(1) English is written in an acceptable form.

(2) The experimental design is easy to understand.

(3) The powders were well characterized using many characterization tools.

(4) Results are in detail and supported by discussion.

Weaknesses:

(1) In the introduction, it is suggested to list examples of the previous approaches to fabricated ZrB2-SiC composite powders and state the innovation/difference of this study compared to them (line 40).

(2) Line 35, any other benefits when SiC whiskers were introduced to the ZrB2 ceramics? The increase in fracture toughness seems not quite a lot (from 3-4 to >5).

(3) Suggest providing the suppliers of the powders so researchers can repeat the work.

(4) Line 68: do you mean ZS0, ZS1, …? Since Table 1 lists ZS0…

(5) Line 139: the present paper form is for publication, so make sure the figure caption is on the same page with the figure. This also works for Figure 6.

(6) It seems there is little results & discussion for Fig. 6c, e, f? Can the figures tell some findings?
